# Minimax Optimal Algorithms for Unconstrained Linear Optimization

**H. Brendan McMahan**
Google Reasearch
Seattle, WA
mcmahan@google.com

**Jacob Abernethy**[*]
Computer Science and Engineering
University of Michigan
jabernet@umich.edu

## Abstract

We design and analyze minimax-optimal algorithms for online linear optimization games where the player's choice is unconstrained. The player strives to minimize regret, the difference between his loss and the loss of a post-hoc benchmark strategy. While the standard benchmark is the loss of the best strategy chosen from a bounded comparator set, we consider a very broad range of benchmark functions. The problem is cast as a sequential multi-stage zero-sum game, and we give a thorough analysis of the minimax behavior of the game, providing characterizations for the value of the game, as well as both the player's and the adversary's optimal strategy. We show how these objects can be computed efficiently under certain circumstances, and by selecting an appropriate benchmark, we construct a novel hedging strategy for an unconstrained betting game.

## 1 Introduction

Minimax analysis has recently been shown to be a powerful tool for the construction of online learning algorithms [Rakhlin et al., 2012]. Generally, these results use bounds on the value of the game (often based on the sequential Rademacher complexity) in order to construct efficient algorithms. In this work, we show that when the learner is unconstrained, it is often possible to efficiently compute an exact minimax strategy for both the player and nature. Moreover, with our tools we can analyze a much broader range of problems than have been previously considered.

We consider a game where on each round $t = 1, \ldots, T$, first the learner selects $x_t \in \mathbb{R}^n$, and then an adversary chooses $g_t \in \mathcal{G} \subset \mathbb{R}^n$, and the learner suffers loss $g_t \cdot x_t$. The goal of the learner is to minimize regret, that is, loss in excess of that achieved by a post-hoc benchmark strategy. We define

$$\text{Regret} = \text{Loss} - (\text{Benchmark Loss}) = \sum_{t=1}^{T} g_t \cdot x_t - L(g_1, \ldots, g_T) \tag{1}$$

as the regret with respect to benchmark performance $L$ (the $L$ intended will be clear from context). The standard definition of regret arises from the choice

$$L(g_1, \ldots, g_T) = \inf_{x \in \mathcal{X}} g_{1:T} \cdot x = \inf_{x \in \mathbb{R}^n} g_{1:T} \cdot x + I(x \in \mathcal{X}), \tag{2}$$

where $I(\texttt{condition})$ is the indicator function: it returns 0 when $\texttt{condition}$ holds, and returns $\infty$ otherwise. The above choice of $L$ represents the loss of the best fixed point $x$ in the bounded convex set $\mathcal{X}$. Throughout we shall write $g_{1:t} = \sum_{s=1}^{t} g_s$ for a sum of scalars or vectors. When $L$ depends only on the sum $G \equiv g_{1:T}$ we write $L(G)$.

---

[*]Work performed while the author was in the CIS Department at the University of Pennsylvania and funded by a Simons Postdoctoral Fellowship

In the present work we shall consider a broad notion of regret in which, for example, $L$ is defined not in terms of a "best in hindsight" comparator but instead in terms of a "penalized best in hindsight" objective. Let $\Psi$ be some penalty function, and consider

$$L(G) = \min_x G \cdot x + \Psi(x). \tag{3}$$

This is a direct generalization of the usual comparator notion which takes $\Psi(x) = I(x \in \mathcal{X})$.

We view this interaction as a sequential zero-sum game played over $T$ rounds, where the player strives to minimize Eq. (1), and the adversary attempts to maximize it. We study the value of this game, defined as

$$V^T \equiv \inf_{x_1 \in \mathbb{R}^n} \sup_{g_1 \in \mathcal{G}} \ldots \inf_{x_T \in \mathbb{R}^n} \sup_{g_T \in \mathcal{G}} \left( \sum_{t=1}^{T} g_t \cdot x_t - L(g_1, \ldots, g_T) \right). \tag{4}$$

With this in mind, we can describe the primary contributions of the present paper:

1. We provide a characterization of the value of the game Eq. (4) in terms of the supremum over the expected value of a function of a martingale difference sequence. This will be made more explicit in Section 2.

2. We provide a method for computing the player's minimax optimal (deterministic) strategy in terms of a "discrete derivative." Similarly, we show how to describe the adversary's optimal randomized strategy in terms of martingale differences.

3. For "coordinate-decomposable" games we give a natural and efficiently computable description of the value of the game and the player's optimal strategy.

4. In Section 3, we consider several benchmark functions $L$, defined in Eq. (3) via a penalty function $\Psi$, which lead to interesting and surprising optimal algorithms; we also exactly compute the values of these games. Figure 1 summarizes these applications. In particular, we show that constant-step-size gradient descent is minimax optimal for a quadratic $\Psi$, and an exponential $L$ leads to a bounded-loss hedging algorithm that can still yield exponential reward on "easy" sequences.

**Applications**    The primary contributions of this paper are to the theory. Nevertheless, it is worth pausing to emphasize that the framework of "unconstrained online optimization" is a fundamental template for (and strongly motivated by) several online learning settings, and the results we develop are applicable to a wide range of commonly studied algorithmic problems. The classic algorithm for linear pattern recognition, the Perceptron, can be seen as an algorithm for unconstrained linear optimization. Methods for training a linear SVM or a logistic regression model, such as stochastic gradient descent or the Pegasos algorithm [Shalev-Shwartz et al., 2011], are unconstrained optimization algorithms. Finally, there has been recent work in the pricing of options and other financial derivatives [DeMarzo et al., 2006, Abernethy et al., 2012] that can be described exactly in terms of a repeated game which fits nicely into our framework.

We also wish to emphasize that the algorithm of Section 3.2 is both practical and easily implementable: for a multi-dimensional problem one needs to only track the sum of gradients for each coordinate (similar to Dual Averaging), and compute Eq. (12) for each coordinate to derive the appropriate strategy. The algorithm provides us with a tool for making potentially unconstrained bets/investments, but as we discuss it also leads to interesting regret bounds.

**Related Work**    Regret-based analysis has received extensive attention in recent years; see Shalev-Shwartz [2012] and Cesa-Bianchi and Lugosi [2006] for an introduction. The analysis of alternative notions of regret is also not new. Vovk [2001] gives bounds relative to benchmarks similar to Eq. (3), though for different problems and not in the minimax setting. In the expert setting, there has been much work on tracking a shifting sequence of experts rather than the single best expert; see Koolen et al. [2012] and references therein. Zinkevich [2003] considers drifting comparators in an online convex optimization framework. This notion can be expressed by an appropriate $L(g_1, \ldots, g_T)$, but now the order of the gradients matters. Merhav et al. [2006] and Dekel et al. [2012] consider the stronger notion of policy regret in the online experts and bandit settings, respectively. Stoltz [2011] also considers some alternative notions of regret. For investing scenarios, Agarwal et al. [2006]

| setting | $L(G)$ | $\Psi(x)$ | minimax value | update |
|---|---|---|---|---|
| soft feasible set | $-\frac{G^2}{2\sigma}$ | $\frac{\sigma}{2}x^2$ | $\frac{T}{2\sigma}$ | $x_{t+1} = \frac{1}{\sigma}g_{1:t}$ |
| standard regret | $-|G|$ | $I(|x| \le 1)$ | $\to \sqrt{\frac{2}{\pi}T}$ | Eq. (14) |
| bounded-loss betting | $-\exp(G/\sqrt{T})$ | $-\sqrt{T}x\log(-\sqrt{T}x) + \sqrt{T}x$ | $\to \sqrt{e}$ | Eq. (12) |

Figure 1: Summary of specific online linear games considered in Section 3. Results are stated for the one-dimensional problem where $g_t \in [-1, 1]$; Corollary 5 gives an extension to $n$ dimensions. The benchmark $L$ is given as a function of $G = g_{1:T}$. The standard notion of regret corresponds to the $L(G) = \min_{x \in [-1,1]} g_{1:t} \cdot x = -|G|$. The benchmark functions can alternatively be derived from a suitable penalty $\Psi$ on comparator points $x$, so $L(G) = \min_x Gx + \Psi(x)$.

and Hazan and Kale [2009] consider regret with respect to the best constant-rebalanced portfolio. Our algorithm in Section 3.2 applies to similar problems, but does not require a "no junk bonds" assumption, and is in fact minimax optimal for a natural benchmark.

Existing algorithms do offer bounds for unconstrained problems, generally of the form $\|x^*\|/\eta + \eta \sum_t g_t x_t$. However, such bounds can only guarantee no-regret when an upper bound $R$ on $\|x^*\|$ is known in advance and used to tune the parameter $\eta$. If one knows such a $R$, however, the problem is no longer truly unconstrained. The only algorithms we know that avoid this problem are those of Streeter and McMahan [2012], and the minimax-optimal algorithm we introduce in Sec 3.2; these algorithms guarantee guarantee $\text{Regret} \le \mathcal{O}\big(R\sqrt{T}\log((1+R)T)\big)$ for any $R > 0$.

The field has seen a number of minimax approaches to online learning. Abernethy and Warmuth [2010] and Abernethy et al. [2008b] give the optimal behavior for several zero-sum games against a budgeted adversary. Section 3.3 studies the online linear game of Abernethy et al. [2008a] under different assumptions, and we adapt some techniques from Abernethy et al. [2009, 2012]; the latter work also involves analyzing an unconstrained player. Rakhlin et al. [2012] utilizes powerful tools for non-constructive analysis of online learning as a technique to design algorithms; our work differs in that we focus on cases where the exact minimax strategy can be computed.

**Notions of Regret** The standard notion of regret corresponds to a hard penalty $\Psi(x) = I(x \in \mathcal{X})$. Such a definition makes sense when the player by definition must select a strategy from some bounded set, for example a probability from the $n$-dimensional simplex, or a distribution on paths in a graph. However, in contexts such as machine learning where any $x \in \mathbb{R}^n$ corresponds to a valid model, such a hard constraint is difficult to justify; while any $x \in \mathbb{R}^n$ is technically feasible, in order to prove regret bounds we compare to a much more restrictive set. As an alternative, in Sections 3.1 and 3.2 we propose soft penalty functions that encode the belief that points near the origin are more likely to be optimal (we can always re-center the problem to match our beliefs in this regard), but do not rule out any $x \in \mathbb{R}^n$ a priori.

Thus, one of our contributions is showing that interesting results can be obtained by choosing $L$ differently than in Eq. (2). The player cannot do well in terms of the absolute loss $\sum_t g_t \cdot x_t$ for all sequences $g_1, \ldots, g_T$, but she can do better on some sequences at the expense of doing worse on others. The benchmark $L$ makes this notion precise: sequences for which $L(g_1, \ldots, g_T)$ is large and negative are those on which the player desires good performance, at the expense of allowing more loss (in absolute terms) on sequences where $L(g_1, \ldots, g_T)$ is large and positive. The value of the game $V^T$ tells us to what extent any online algorithm can hope to match the benchmark $L$.

## 2 General Unconstrained Linear Optimization

In this section we develop general results on the unconstrained linear optimization problem. We start by analyzing (4) in greater detail, and give tools for computing the regret value $V^T$ in such games. We show that in certain cases the computation of the minimax value can be greatly simplified.

Throughout we will assume that the function $L$ is concave in each of its arguments (thought not necessarily jointly concave) and bounded on $\mathcal{G}^T$. We also include the following assumptions on the

set $\mathcal{G}$. First, we assume that either $\mathcal{G}$ is a polytope or, more generally, that ConvexHull$(\mathcal{G})$ is a full-rank polytope in $\mathbb{R}^n$. This is not strictly necessary but is convenient for the analysis; any bounded convex set in $\mathbb{R}^n$ can be approximated to arbitrary precision with a polytope. We also make the necessary assumption that the ConvexHull$(\mathcal{G})$ contains the origin in its interior. We let $\mathcal{G}'$ be the set of "corners" of $\mathcal{G}$, that is $\mathcal{G}' = \{g^1, \ldots, g^m\}$ and hence ConvexHull$(G)$ = ConvexHull$(\mathcal{G}')$.

We are also concerned with the conditional value of the game, $V_t$, given $x_1, \ldots x_t$ and $g_1, \ldots g_t$ have already been played. That is, the Regret when we fix the plays on the first $t$ rounds, and then assume minimax optimal play for rounds $t+1$ through $T$. However, following the approach of Rakhlin et al. [2012], we omit the terms $\sum_{s=1}^{t} x_s \cdot g_s$ from Eq. (4). We can view this as cost that the learner has already payed, and neither that cost nor the specific previous plays of the learner impact the value of the remaining terms in Eq. (1). Thus, we define

$$V_t(g_1, \ldots, g_t) = \inf_{x_{t+1} \in \mathbb{R}^n} \sup_{g_{t+1} \in \mathcal{G}} \ldots \inf_{x_T \in \mathbb{R}^n} \sup_{g_T \in \mathcal{G}} \left( \sum_{s=t+1}^{T} g_s \cdot x_s - L(g_1, \ldots, g_T) \right). \quad (5)$$

Note the conditional value of the game before anything has been played, $V_0()$, is exactly $V^T$.

**The martingale characterization of the game**   The fundamental tool used in the rest of the paper is the following characterization of the conditional value of the game:

**Theorem 1.** *For every $t$ and every sequence $g_1, \ldots, g_t \in \mathcal{G}$, we can write the conditional value of the game as*

$$V_t(g_1, \ldots, g_t) = \max_{G \in \Delta(\mathcal{G}'), \mathbb{E}[G]=0} \mathbb{E}[V_{t+1}(g_1, \ldots, g_t, G)],$$

*where $\Delta(\mathcal{G}')$ is the set of random variables on $\mathcal{G}'$. Moreover, for all $t$ the function $V_t$ is convex in each of its coordinates and bounded.*

All proofs omitted from the body of the paper can be found in the appendix or the extended version of this paper.

Let $M_T(\mathcal{G})$ be the set of $T$-length *martingale difference sequences* on $\mathcal{G}'$, that is the set of all sequences of random variables $(G_1, \ldots, G_T)$, with $G_t$ taking values in $\mathcal{G}'$, which satisfy $\mathbb{E}[G_t | G_1, \ldots, G_{t-1}] = 0$ for all $t = 1, \ldots, T$. Then, we immediately have the following:

**Corollary 2.** *We can write*

$$V^T = \max_{(G_1, \ldots, G_T) \in M_T(\mathcal{G}')} \mathbb{E}[-L(G_1, \ldots, G_T)],$$

*with the analogous expression holding for the conditional value of the game.*

**Characterization of optimal strategies**   The result above gives a nice expression for the value of the game $V^T$ but unfortunately it does not lead directly to a strategy for the player. We now dig a bit deeper and produce a characterization of the optimal player behavior. This is achieved by analyzing a simple one-round zero-sum game. As before, we assume $\mathcal{G}$ is a bounded subset of $\mathbb{R}^n$ whose convex hull is a polytope whose interior contains the the origin $\mathbf{0}$. Assume we are given some convex function $f$ defined and bounded on all of ConvexHull$(\mathcal{G})$. We consider the following:

$$V = \inf_{x \in \mathbb{R}^n} \sup_{g \in \mathcal{G}} x \cdot g + f(g). \quad (6)$$

**Theorem 3.** *There exists a set of $n+1$ distinct points $\{g^1, \ldots, g^{n+1}\} \subset \mathcal{G}$ whose convex hull is of full rank, and a distribution $\vec{\alpha} \in \Delta_{n+1}$ satisfying $\sum_{i=1}^{n+1} \alpha_i g^i = \mathbf{0}$, such that $V = \sum_{i=1}^{n+1} \alpha_i f(g^i)$. Moreover, an optimal choice for the infimum in (6) is the gradient of the unique linear interpolation of the pairs $\{(g^1, -f(g^1)), \ldots, (g^{n+1}, -f(g^{n+1}))\}$.*

The theorem makes a useful point about determining the player's optimal strategy for games of this form. If the player can determine a full-rank set of "best responses" $\{g^1, \ldots, g^{n+1}\}$ to his optimal $x^*$, each of which should be a corner of the polytope $\mathcal{G}$, then we know that $x^*$ must be a "discrete gradient" of the function $-f$ around $\mathbf{0}$. That is, if the size of $\mathcal{G}$ is small relative to the curvature of $f$, then an approximation to $-\nabla f(\mathbf{0})$ is the linear interpolation of $-f$ at a set of points around $\mathbf{0}$. An optimal $x^*$ will be exactly this interpolation.

This result also tells us how to analyze the general $T$-round game. We can express (5), the conditional value of the game $V_{t-1}$, in recursive form as

$$V_{t-1}(g_1, \ldots, g_{t-1}) = \inf_{x_t \in \mathbb{R}^n} \sup_{g_t \in \mathcal{G}} g_t \cdot x_t + V_t(g_1, \ldots, g_{t-1}, g_t). \tag{7}$$

Hence by setting $f(g_t) = V_t(g_1, \ldots, g_{t-1}, g_t)$, noting that the latter is convex in $g_t$ by Theorem 1, we see we have an immediate use of Theorem 3.

## 3 Minimax Optimal Algorithms for Coordinate-Decomposable Games

In this section, we consider games where $\mathcal{G}$ consists of axis-aligned constraints, and $L$ decomposes so $L(g) = \sum_{i=1}^n L_i(g_i)$. In order to solve such games, it is generally sufficient to consider $n$ independent one-dimensional problems. We study such games first:

**Theorem 4.** *Consider the one-dimensional unconstrained game where the player selects $x_t \in \mathbb{R}$ and the adversary chooses $g_t \in \mathcal{G} = [-1, 1]$, and $L$ is concave in each of its arguments and bounded on $\mathcal{G}^T$. Then, $V^T = \mathbb{E}_{g_t \sim \{-1,1\}} \big[ -L(g_1, \ldots, g_T) \big]$ where the expectation is over each $g_t$ chosen independently and uniformly from $\{-1, 1\}$ (that is, the $g_t$ are Rademacher random variables). Further, the conditional value of the game is*

$$V_t(g_1, \ldots, g_t) = \mathbb{E}_{g_{t+1}, \ldots, g_T \sim \{-1,1\}} \big[ -L(g_1, \ldots, g_t, g_{t+1}, \ldots g_T) \big]. \tag{8}$$

The proof is immediate from Corollary 2, since the *only* possible martingale that both plays from the corners of $\mathcal{G}$ and has expectation 0 on each round is the sequence of independent Rademacher random variables.[1] Given Theorem 4, and the fact that the functions $L$ of interest will generally depend only on $g_{1:T}$, it will be useful to define $\mathcal{B}_T$ to be the distribution of $g_{1:T}$ when each $g_t$ is drawn independently and uniformly from $\{-1, 1\}$.

Theorem 4 can immediately be extended to coordinate-decomposable games as follows:

**Corollary 5.** *Consider the game where the player chooses $x_t \in \mathbb{R}^n$, the adversary chooses $g_t \in [-1, 1]^n$, and the payoff is $\sum_{t=1}^T g_t \cdot x_t - \sum_{i=1}^n L(g_{1:T,i})$ for concave L. Then the value $V^T$ and the conditional value $V_t(\cdot)$ can be written as*

$$V^T = n \mathbb{E}_{G \sim \mathcal{B}_T} \big[ -L(G) \big] \quad and \quad V_t(g_1, \ldots, g_t) = \sum_{i=1}^n \mathbb{E}_{G_i \sim \mathcal{B}_{T-t}} \big[ -L(g_{1:t,i} + G_i) \big].$$

The proof follows by noting the constraints on both players' strategies and the value of the game fully decompose on a per-coordinate basis.

**A recipe for minimax optimal algorithms in one dimension** Since Eq. (5) gives the minimax value of the game if both players play optimally from round $t + 1$ forward, a minimax strategy for the learner on round $t + 1$ must be $x_{t+1} = \arg\min_{x \in \mathbb{R}} \max_{g \in \{-1,1\}} g \cdot x + V_{t+1}(g_1, \ldots, g_t, g)$. Now, we can apply Theorem 3, and note that unique strategy for the adversary is to play $g = 1$ or $g = -1$ with equal probability. Thus, the player strategy is just the interpolation of the points $(-1, -f(-1))$ and $(1, -f(1))$, where we take $f = V_{t+1}$, giving us

$$x_{t+1} = \frac{1}{2} \big( V_{t+1}(g_1, \ldots, g_t, -1) - V_{t+1}(g_1, \ldots, g_t, +1) \big). \tag{9}$$

Thus, if we can derive a closed form for $V_t(g_1, \ldots, g_t)$, we will have an efficient minimax-optimal algorithm. Note that for any function $L$,

$$\mathbb{E}_{G \sim \mathcal{B}_T}[L(G)] = \frac{1}{2^T} \sum_{i=0}^T \binom{T}{i} L(2i - T), \tag{10}$$

since $2^{-T} \binom{T}{i}$ is the binomial probability of getting exactly $i$ gradients of $+1$ over $T$ rounds, which implies $T - i$ gradients of $-1$, so $G = i - (T - i) = 2i - T$. Using Theorem 4, and Eqs (9) and (10), in

the following sections we exactly compute the game values and unique minimax optimal strategies for a variety of interesting coordinate-decomposable games. Even when such exact computations are not possible, *any* coordinate-decomposable game where $L$ depends only on $G = g_{1:T}$ can be solved numerically in polynomial time. If $\tau = T - t$, the number of rounds remaining, then we can compute $V_t$ exactly by using the appropriate binomial probabilities (following Eq. (8) and Eq. (10)), requiring only a sum over $\mathcal{O}(\tau)$ values. If $\tau$ is large enough, then using an approximation to the binomial (e.g., the Gaussian approximation) may be sufficient.

We can also immediately provide a characterization of the potentially optimal player strategies in terms of the subgradients of $-L$. For simplicity, we write $-\partial L(g)$ instead of $\partial(-L(g))$.

**Theorem 6.** *Let $\mathcal{G} = [a, b]$, with $a < 0 < b$, and $L : \mathbb{R} \to \mathbb{R}$ is bounded and concave. Then, on every round, the unique minimax optimal $x_t^*$ satisfies $-x_t^* \in \mathcal{L}$ where $\mathcal{L} = \cup_{w \in \mathbb{R}} - \partial L(w)$.*

*Proof.* Following Theorem 3, we know the minimax $x_{t+1}$ interpolates $(a, -f(a))$ and $(b, -f(b))$, where we take $f(g) = V_{t+1}(g_1, \ldots, g_t, g)$. In one dimension, this implies $-x_{t+1} \in \partial f(g)$ for some $g \in \mathcal{G}$. It remains to show $\partial f(g) \subseteq \mathcal{L}$. From Theorem 1 we have $f(g) = \mathbb{E}[-L(g_{1:t} + g + B)]$, where the $\mathbb{E}$ is with respect to mean-zero random variable $B \sim \mathcal{B}_\tau$, $\tau = T - t$. For each possible value $b$ that $B$ can take on, $-\partial_g L(g_{1:t} + g + b_i) \subseteq \mathcal{L}$ by definition, so $\partial f(g)$ is a convex combination of these sets (e.g., Rockafellar [1997, Thm. 23.8]). The result follows as $\mathcal{L}$ is convex. $\square$

Note that for standard regret, $L(g) = \inf_{x \in \mathcal{X}} gx$, we have $\partial L(g) \subseteq \mathcal{X}$, indicating that (in 1 dimension at least), the player never needs to play outside the comparator set $\mathcal{X}$. We will see additional consequences of this theorem in the following sections.

## 3.1 Constant step-size gradient descent can be minimax optimal

Suppose we use a "soft" feasible set for the benchmark via a quadratic penalty,

$$L(G) = \min_x \ Gx + \frac{\sigma}{2}x^2 = -\frac{1}{2\sigma}G^2, \tag{11}$$

for a constant $\sigma > 0$. Does a no-regret algorithm against this comparison class exist? Unfortunately, the general answer is no, as shown in the next theorem. Recalling $g_t \in [-1, 1]$,

**Theorem 7.** *The value of this game is $V^T = \mathbb{E}_{G \sim \mathcal{B}_T}\left[\frac{1}{2\sigma}G^2\right] = \frac{T}{2\sigma}$.*

Thus, for a fixed $\sigma$, we cannot have a no regret algorithm with respect to this $L$. But this does not mean the minimax algorithm will be uninteresting. To derive the minimax optimal algorithm, we compute conditional values (using similar techniques to Theorem 7),

$$V_t(g_1, \ldots, g_t) = \mathop{\mathbb{E}}_{G \sim \mathcal{B}_{T-t}}\left[\frac{1}{2\sigma}(g_{1:t} + G)^2\right] = \frac{1}{2\sigma}\left((g_{1:t})^2 + (T - t)\right),$$

and so following Eq. (9) the minimax-optimal algorithm must use

$$x_{t+1} = \frac{1}{4\sigma}\left(((g_{1:t} - 1)^2 + (T - t - 1)) - ((g_{1:t} + 1)^2 + (T - t - 1))\right) = \frac{1}{4\sigma}(-4g_{1:t}) = -\frac{1}{\sigma}g_{1:t}$$

Thus, a minimax-optimal algorithm is simply constant-learning-rate gradient descent with learning rate $\frac{1}{\sigma}$. Note that for a fixed $\sigma$, this is the optimal algorithm independent of $T$; this is atypical, as usually the minimax optimal algorithm depends on the horizon (as we will see in the next two cases). Note that the set $\mathcal{L} = \mathbb{R}$ (from Theorem 6), and indeed the player could eventually play an arbitrary point in $\mathbb{R}$ (given large enough $T$).

## 3.2 Non-stochastic betting with exponential upside and bounded worst-case loss

A major advantage of the regret minimization framework is that the guarantees we can achieve are typically robust to arbitrary input sequences. But on the downside the model is very pessimistic: we measure performance *in the worst case*. One might aim to perform not too badly in the worst case yet extremely well under certain conditions.

We now show how the results in the present paper can lead to a very optimistic guarantee, particularly in the case of a sequential betting game. On each round $t$, the world offers the player a betting opportunity on a coin toss, i.e. a binary outcome $g_t \in \{-1, 1\}$. The player may take either side of the bet, and selects a wager amount $x_t$, where $x_t > 0$ implies a bet on tails ($g_t = -1$) and $x_t < 0$ a bet on heads ($g_t = 1$). The world then announces whether the bet was won or lost, revealing $g_t$. The player's wealth changes (additively) by $-g_t x_t$ (that is, the player strives to minimize loss $g_t x_t$). We assume that the player begins with some initial capital $\alpha > 0$, and at any time period the wager $|x_t|$ must not exceed $\alpha - \sum_{s=1}^{t-1} g_s x_s$, the initial capital plus the money earned thus far.

With the benefit of hindsight, the gambler can see $G = \sum_{t=1}^{T} g_t$, the total number of heads minus the total number of heads. Let us imagine that the number of heads significantly exceeded the number of tails, or vice versa; that is, $|G|$ was much larger than 0. Without loss of generality let us assume that $G$ is positive. Let us imagine that the gambler, with the benefit of hindsight, considers what could have happened had he always bet a constant fraction $\beta$ of his wealth on heads. A simple exercise shows that his wealth would become

$$\prod_{t=1}^{T}(1 + \beta g_t) = (1 + \beta)^{\frac{T+G}{2}}(1 - \beta)^{\frac{T-G}{2}}.$$

This is optimized at $\beta = \frac{G}{T}$, which gives a simple expression in terms of KL-divergence for the maximum wealth in hindsight, $\exp\left(T \cdot \text{KL}\left(\frac{1+G/T}{2} \,||\, \frac{1}{2}\right)\right)$, and the former is well-approximated by $\exp(O(G^2/T))$ when $G$ is not too large relative to $T$. In other words, with knowledge of the final $G$, a naïve betting strategy could have earned the gambler *exponentially large* winnings starting with constant capital. Note that this is essentially a Kelly betting scheme [Kelly Jr, 1956], expressed in terms of $G$. We ask: does there exist an adaptive betting strategy that can compete with this hindsight benchmark, even if the $g_t$ are chosen fully adversarially?

Indeed we show we can get reasonably close. Our aim will be to compete with a slightly weaker benchmark $L(G) = -\exp(|G|/\sqrt{T})$. We present a solution for the one-sided game, without the absolute value, so the player only aims for exponential wealth growth for large positive $G$. It is not hard to develop a two-sided algorithm as a result, which we soon discuss.

**Theorem 8.** *Consider the game where $\mathcal{G} = [-1, 1]$ with benchmark $L(G) = -\exp(G/\sqrt{T})$. Then*

$$V^T = \left(\cosh \frac{1}{\sqrt{T}}\right)^T \leq \sqrt{e}$$

*with the bound tight as $T \to \infty$. Let $\tau = T - t$ and $G_t = g_{1:t}$, then the conditional value of the game is $V_t(G_t) = \left(\cosh \frac{1}{\sqrt{T}}\right)^\tau \exp\left(\frac{G_t}{\sqrt{T}}\right)$ and the player's minimax optimal strategy is:*

$$x_{t+1} = -\exp\left(\frac{G_t}{\sqrt{T}}\right)\sinh\frac{1}{\sqrt{T}}\left(\cosh\frac{1}{\sqrt{T}}\right)^{\tau-1} \tag{12}$$

Recall that the value of the game can be thought of as the largest possible difference between the payoff of the benchmark function $\exp(G/\sqrt{T})$ and the winnings of the player $-\sum g_t x_t$, when the player uses an optimal betting strategy. That the value of the game here is of constant order is critical, since it says that we can always achieve a payoff that is exponential in $\frac{G}{\sqrt{T}}$ at a cost of no more than $\sqrt{e} = O(1)$. Notice we have said nothing thus far regarding the nature of our betting strategy; in particular we have not proved that the strategy satisfies the required condition that the gambler cannot bet more than $\alpha$ plus the earnings thus far. We now give a general result showing that this condition can be satisfied:

**Theorem 9.** *Consider a one dimensional game with $\mathcal{G} = [-1, 1]$ with benchmark function $L$ non-positive on $\mathcal{G}^T$. Then for the optimal betting strategy we have that $|x_t| \leq -\sum_{s=1}^{t} g_s x_s + V^T$, and further $V^T \geq \sum_{s=1}^{t} g_s x_s$ for any $t$ and any sequence $g_1, \ldots, g_t$.*

In other words, the player's cumulative loss at any time is always bounded from below by $V^T$. This implies that the starting capital $\alpha$ required to "replicate" the payoff function is *exactly* the value[2] of the game $V^T$. Indeed, to replicate $\exp(G/\sqrt{T})$ we would require no more than $\alpha = \$1.65$.

It is worth noting an alternative characterization of the benchmark function $L$ used here. For $a \geq 0$, $\min_{x \in \mathbb{R}^-} (Gx - ax \log(-ax) + ax) = -\exp\left(\frac{G}{a}\right)$. Thus, if we take $\Psi(x) = -ax \log(ax) + ax + I(x \leq 0)$, we have $\min_{x \in \mathbb{R}^-} g_{1:T} x + \Psi(x) = -\exp\left(\frac{G}{a}\right)$. Since this algorithm needs large Reward when $G$ is large and positive, we might expect that the minimax optimal algorithm only plays $x_t \leq 0$. Another intuition for this is that the algorithm should not need to play any point $x$ to which $\Psi$ assigns an infinite penalty. This intuition can be confirmed immediately via Theorem 6.

We now sketch how to derive an algorithm for the "two-sided" game. To do this, we let $L_C(G) \equiv L(G) + L(-G) \leq -\exp(|G|/\sqrt{T})$. We can construct a minimax optimal algorithm for $L_C(G)$ by running two copies of the one-sided minimax algorithm simultaneously, switching the signs of the gradients and plays of the second copy. We formalize this in Appendix B.

This same benchmark and algorithm can be used in the setting introduced by Streeter and McMahan [2012]. In that work, the goal was to prove bounds on standard regret like $\text{Regret} \leq \mathcal{O}(R\sqrt{T} \log((1+R)T))$ simultaneously for any comparator $x^*$ with $|x^*| = R$. Stating their Theorem 1 in terms of losses, this traditional regret bound is achieved by any algorithm that guarantees

$$\text{Loss} = \sum_{t=1}^{T} g_t x_t \leq -\exp\left(\frac{|G|}{\sqrt{T}}\right) + \mathcal{O}(1). \tag{13}$$

The symmetric algorithm (Appendix B) satisfies

$$\text{Loss} \leq -\exp\left(\frac{G}{\sqrt{T}}\right) - \exp\left(\frac{-G}{\sqrt{T}}\right) + 2\sqrt{e} \leq -\exp\left(\frac{|G|}{\sqrt{T}}\right) + 2\sqrt{e},$$

and so we also achieve a standard regret bound of the form given above.

### 3.3  Optimal regret against hypercube adversaries

Perhaps the simplest and best studied learning games are those that restrict both the player and adversary to a norm ball, and use the standard notion of regret. We can derive results for the game where the adversary has an $L_\infty$ constraint, the comparator set is also the $L_\infty$ ball, and the player is unconstrained. Corollary 5 implies it is sufficient to study the one-dimensional case.

**Theorem 10.** *Consider the game between an adversary who chooses losses $g_t \in [-1, 1]$, and a player who chooses $x_t \in \mathbb{R}$. For a given sequence of plays, $x_1, g_1, x_2, g_2, \ldots, x_T, g_T$, the value to the adversary is $\sum_{t=1}^{T} g_t x_t - |g_{1:T}|$. Then, when $T$ is even with $T = 2M$, the minimax value of this game is given by*

$$V_T = 2^{-T} \frac{2M\, T!}{(T-M)! M!} \leq \sqrt{\frac{2T}{\pi}}.$$

*Further, as $T \to \infty$, $V_T \to \sqrt{\frac{2T}{\pi}}$. Let $B$ be a random variable drawn from $\mathcal{B}_{T-t}$. Then the minimax optimal strategy for the player given the adversary has played $G_t = g_{1:t}$ is given by*

$$x_{t+1} = \Pr(B < -G_t) - \Pr(B > -G_t) = 1 - 2\Pr(B > -G_t) \in [-1, 1]. \tag{14}$$

The fact that the limiting value of this game is $\sqrt{2T/\pi}$ was previously known, e.g., see a mention in Abernethy et al. [2009]; however, we believe this explicit form for the optimal player strategy is new. This strategy can be efficiently computed numerically, e.g, by using the regularized incomplete beta function for the CDF of the binomial distribution. It also follows from this expression that even though we allow the player to select $x_{t+1} \in \mathbb{R}$, the minimax optimal algorithm always selects points from $[-1, 1]$, so our result applies to the case where the player is constrained to play from $\mathcal{X}$.

Abernethy et al. [2008a] shows that for the linear game with $n \geq 3$ where both the learner and adversary select vectors from the unit sphere, the minimax value is exactly $\sqrt{T}$. Interestingly, in the $n = 1$ case (where $L_2$ and $L_\infty$ coincide), the value of the game is lower, about $0.8\sqrt{T}$ rather than $\sqrt{T}$. This indicates a fundamental difference in the geometry of the $n = 1$ space and $n \geq 3$. We conjecture the minimax value for the $L_2$ game with $n = 2$ lies somewhere in between.

## Footnotes

[1] However, is easy to extend this to the case where $\mathcal{G} = [a, b]$, which leads to different random variables.

[2]This idea has a long history in finance and was a key tool in Abernethy et al. [2012], DeMarzo et al. [2006], and other works.

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
