[Supplementary Material · unconstrained_minimax_nips_appendix.pdf]

# A  Proofs

We restate the results proved here for convenience.

## A.1   Proof of Theorem 3

**Theorem 1.** *For every $t$ and every sequence $g_1, \ldots, g_t \in \mathcal{G}$, we can write the conditional value of the game as*

$$V_t(g_1, \ldots, g_t) = \max_{G \in \Delta(\mathcal{G}'), \mathbb{E}[G]=0} \mathbb{E}[V_{t+1}(g_1, \ldots, g_t, G)],$$

*where $\Delta(\mathcal{G}')$ is the set of random variables on $\mathcal{G}'$. Moreover, for all $t$ the function $V_t$ is convex in each of its coordinates and bounded.*

*Proof.* We prove both statements simultaneously via induction on $t$. For the base case, $t = T - 1$, we have

$$V_{T-1}(g_1, \ldots, g_{T-1}) = \inf_{x_T} \sup_{g_T} g_T \cdot x_T - L(g_1, \ldots, g_{T-1}, g_T).$$

Because the supremum is taken over $\mathcal{G}$ whose convex hull is assumed to be a polytope, we can replace the $\sup_{g_T \in \mathcal{G}}$ with $\max_{g_T \in \mathcal{G}'}$. Furthermore, we can replace the maximization over points from $\mathcal{G}'$ with the maximization of distributions over $\mathcal{G}' = \{g^i\}_{i=1,\ldots,m}$. That is, we can write

$$V_{T-1}(g_1, \ldots, g_{T-1}) = \inf_{x_T} \max_{\vec{\alpha} \in \Delta_m} \sum_{i=1}^{m} \alpha_i(g^i \cdot x_T - L(g_1, \ldots, g_{T-1}, g^i)).$$

The set $\Delta_m$ is a compact convex set, and the objective $\sum_{i=1}^{m} \alpha_i(g^i \cdot x_T - L(g_1, \ldots, g_{T-1}, g^i))$ is linear in both $x$ and $\vec{\alpha}$ hence we can apply Sion's Minimax theorem to obtain

$$V_{T-1}(g_1, \ldots, g_{T-1}) = \max_{\vec{\alpha} \in \Delta_m} \inf_{x_T} \left( \sum_i \alpha_i g^i \right) \cdot x_T - \sum_i \alpha_i L(g_1, \ldots, g_{T-1}, g^i).$$

Notice that if $\sum_i \alpha_i g^i \neq \mathbf{0}$ then the infimum is $-\infty$ since the player can make the objective arbitrarily small. Hence we can restrict the outer maximization to distributions $\vec{\alpha}$ such that $\sum_i \alpha_i g^i = \mathbf{0}$. This simplifies the expression to

$$V_{T-1}(g_1, \ldots, g_{T-1}) = \max_{\vec{\alpha} \in \Delta_m} - \sum_i \alpha_i L(g_1, \ldots, g_{T-1}, g^i) \qquad \text{s.t.} \quad \sum_i \alpha_i g^i = \mathbf{0}.$$

Notice that, by assumption, $-L$ is convex in each of its arguments, and hence $V_{T-1}(g_1, \ldots, g_{T-1})$ is also convex in each $g_t$ independently, since the maximum of convex functions is convex.

The inductive argument follows identically to the base case, but where we replace $-L$ with $V_t$, since we can write

$$V_{t-1}(g_1, \ldots, g_{t-1}) = \inf_{x_t} \sup_{g_t \in \mathcal{G}} g_t \cdot x_t + V_t(g_1, \ldots, g_{t-1}, g_t).$$

$\square$

**Theorem 3.** *There exists a set of $n + 1$ distinct points $\{g^1, \ldots, g^{n+1}\} \subset \mathcal{G}$ whose convex hull is of full rank, and a distribution $\vec{\alpha} \in \Delta_{n+1}$ satisfying $\sum_{i=1}^{n+1} \alpha_i g^i = \mathbf{0}$, such that $V = \sum_{i=1}^{n+1} \alpha_i f(g^i)$. Moreover, an optimal choice for the infimum in (6) is the gradient of the unique linear interpolation of the pairs $\{(g^1, -f(g^1)), \ldots, (g^{n+1}, -f(g^{n+1}))\}$.*

We prove this theorem via a sequence of lemmas. We begin with the observation that we may assume, without loss of generality, that $\mathcal{G}$ is convex, and hence $\mathcal{G} = \text{ConvexHull}(\mathcal{G})$. This is because, for any $x$, the objective $\sup_{g \in \mathcal{G}} x \cdot g + f(g)$ will always be achieved at the boundary of $\mathcal{G}$ since the objective function $x \cdot g + f(g)$ is the sum of two convex functions and is thus convex.

**Lemma 11.** *The infimum in (6) is achieved in a bounded set.*

*Proof.* Let $M = \sup_{g \in \mathcal{G}} |f(g)|$ then clearly we have that $\inf_{x \in \mathbb{R}^n} \sup_{g \in \mathcal{G}} x \cdot g + f(g) \le M$ since $x$ can be chosen as $\mathbf{0}$. It is sufficient to show any $x$ such that $\|x\| > 2M/\epsilon$ achieves a worse value than $\mathbf{0}$. Since $\mathbf{0}$ is in the interior of $\mathcal{G}$, there exists an $\epsilon > 0$ such that $g = \frac{\epsilon x}{\|x\|} \in \mathcal{G}$. Then, $\sup_{g \in \mathcal{G}} x \cdot g + f(g) \ge x \cdot \frac{\epsilon x}{\|x\|} + f(g) > 2M - M = M$. $\square$

The above lemma is useful since it lets us conclude that we need not necessarily assume $x$ is un-bounded. Moreover, since the $\inf$ is achieved on a compact set, then it has at least one solution $x^*$ that we can analyze. Let $\Phi \subset \mathbb{R}^n$ denote the set of points $x$ on which the infimum in (6) is achieved. For any $x$, let $\Gamma(x) \subset \mathcal{G}$ be the set of corners of the polytope $\mathcal{G}$ on which the supremum $\sup_{g \in \mathcal{G}} x \cdot g + f(g)$ is achieved for fixed $x$.

**Lemma 12.** *For any $x \in \Phi$, the set* $\text{ConvexHull}(\Gamma(x))$ *must contain the origin.*

*Proof.* Let us assume that $\mathbf{0} \notin \text{ConvexHull}(\Gamma(x))$, then I will show that this contradicts the assumption that $x$ is optimal. If $v$ is the value of the objective in (6), then define $\Gamma_\epsilon(x)$ to be the set of $g \in \mathcal{G}$ such that $g \cdot x + f(x) \ge v - \epsilon$. We claim that we can choose $\epsilon > 0$ small enough so that $\text{ConvexHull}(\Gamma_\epsilon(x))$ also does not contain $\mathbf{0}$. This implies that there is some $\delta > 0$ such that $\|g\| > \delta$ for all $g \in \text{ConvexHull}(\Gamma_\epsilon(x))$. Moreover, since $\text{ConvexHull}(\Gamma_\epsilon(x))$ is a convex set there must be a separating hyperplane between $\mathbf{0}$ and $\text{ConvexHull}(\Gamma_\epsilon(x))$, and hence there is some unit vector $z \in \mathbb{R}^n$ (the normal to the hyperplane) such that $z \cdot g < -\delta$ for all $g \in \text{ConvexHull}(\Gamma_\epsilon(x))$.

Choose $B > 0$ such that $\|g\| \le B$ for all $g \in \mathcal{G}$. We claim that the point $x' \equiv x + \frac{\epsilon}{2B} z$ has a strictly smaller objective value that $x$. Consider any $g \in \text{ConvexHull}(\Gamma_\epsilon(x))$, then we have

$$g \cdot x' + f(g) = g \cdot x + f(g) + \frac{\epsilon}{2B} z \cdot g < v - \frac{\epsilon \delta}{2B} < v.$$

On the other hand, for any $g \in \mathcal{G} \setminus \text{ConvexHull}(\Gamma_\epsilon(x))$ we have

$$g \cdot x' + f(g) = g \cdot x + f(g) + \frac{\epsilon}{2B} z < v - \epsilon + \frac{\epsilon}{2B} z \cdot g \le v - \epsilon + \frac{\epsilon}{2B} \|g\| \le v - \frac{\epsilon}{2} < v$$

where the first inequality follows because by assumption $g \notin \Gamma_\epsilon(x)$. It follows from these two expressions that $\sup_{g \in G} g \cdot x' + f(g) < \sup_{g \in G} g \cdot x + f(g)$, a contradiction. $\square$

Concluding that $\text{ConvexHull}(H)$ contains the origin is actually surprisingly useful.

**Lemma 13.** *There is some $x \in \Phi$ such that* $\text{ConvexHull}(\Gamma(x))$ *has a non-empty interior.*

Another way to put this is that $\Gamma(x)$ has at least $n + 1$ points such that none of these is a convex combination of the others.

*Proof.* Notice that $\Phi$ is a convex set and, via Lemma 11, is bounded and compact. We claim that any $x$ on the boundary of $\Phi$ satisfies the goal of the lemma. Choose a boundary point $x \in \Phi$, and assume that $\text{ConvexHull}(\Gamma(x))$ is not of full-rank. Via Lemma 12, this set contains the origin, and hence we can find some unit vector $z$ such that $z \cdot g = 0$ for all $g \in \text{ConvexHull}(\Gamma(x))$.

Since $\mathcal{G}$ is a polytope, we can describe it as the hull of a finite number of points $\mathcal{G}' \equiv \{g^1, \ldots, g^m\}$. For any $g^i \notin \Gamma(x)$ we have $g^i \cdot x + f(g^i) < v$. Choose some $\epsilon > 0$ so that $g^i \cdot x + f(g^i) < v - \epsilon$ for every $g^i \in \mathcal{G}' \setminus \Gamma(x)$, which is possible since this is a finite set. Let $B > 0$ be a bound on the norm of all points in $\mathcal{G}$. Then we claim that the points $x + \frac{\epsilon}{2B} z$ and $x - \frac{\epsilon}{2B} z$ are both members of $\Phi$. Of course, the latter statement contradicts the assumption that $x$ is at the boundary of $\Phi$. To prove the final claim, notice that by the convexity of $f$ we have

$$\sup_{g \in \mathcal{G}} g \cdot \left( x + \frac{\epsilon}{2B} z \right) + f(g) = \max_{i=1,\ldots,m} g^i \cdot \left( x + \frac{\epsilon}{2B} z \right) + f(g^i).$$

For the last expression, we can check two cases. If $g^i \in \Gamma(x)$ then $g^i \cdot z = 0$ in which case $g^i \cdot \left( x + \frac{\epsilon}{2B} z \right) + f(g^i) = g^i \cdot x + f(g^i)$. On the other hand, for $g^i \notin \Gamma(x)$ we have

$$g^i \cdot \left( x + \frac{\epsilon}{2B} z \right) + f(g^i) = g^i \cdot x + f(g^i) = \frac{\epsilon}{2B} g \cdot z < v - \epsilon + \epsilon/2 < v.$$

Hence the value of the objective is the same for $x$ and $x + \frac{\epsilon}{2B} z$. A similar argument follows for $x - \frac{\epsilon}{2B} z$. $\square$

**Lemma 14.** *If $x \in \Phi$ and we pick any full-rank set of points $g_1, \ldots, g_{n+1} \in \Gamma(x)$ whose hull contains the origin, then we may write $x$ as the gradient of the linear interpolation of the points $\{(g_1, -f(g_1)), \ldots, (g_{n+1}, -f(g_{n+1}))\}$. Moreover, this implies that $x$ is a subgradient of the function $f$ restricted to the set $\mathcal{G}$.*

*Proof.* Let us notice that if we were to search for the linear interpolation of the points $\{(g_1, -f(g_1)), \ldots, (g_{n+1}, -f(g_{n+1}))\}$, then we would need to find a vector $m \in \mathbb{R}^n$ and an off-set $b \in \mathbb{R}$ such that

$$m \cdot g_i + b = -f(g_i) \qquad \forall\, i = 1, \ldots, n+1,$$

and indeed since the set of $g_i$'s is of full rank this has a unique solution. However, the point $x$ also satisfies a similar set of equations:

$$x \cdot g_i + f(g_i) = c \qquad \forall\, i = 1, \ldots, n+1,$$

where $c$ is the value of the objective in (6). Given the uniqueness of the above to systems of equations, we have that $m = x$. $\qquad\square$

Now given the above results we can actually construct the optimal strategy for the adversary.

**Lemma 15.** *For any full-rank set of points $g_1, \ldots, g_{n+1} \in \Gamma(x)$ whose hull contains the origin, let $\vec{\alpha} \in \Delta_{n+1}$ be a set of weights such that $\sum_i \alpha_i g_i = \mathbf{0}$ (and indeed $\vec{\alpha}$ is unique). Then the value of the objective (6) is precisely $\sum_i \alpha_i f(g_i)$. Moreover, one optimal randomized strategy for the adversary is to choose $g_i$ with probability $\alpha_i$.*

*Proof.* Recall that the point $x^*$ satisfies a system of linear equations

$$x^* \cdot g_i + f(g_i) = c \qquad \forall\, i = 1, \ldots, n+1,$$

where $c$ is the value of the objective. Furthermore, it also satisfies any *mixture* of these equations. By taking an $\vec{\alpha}$ mixture of these equations we have

$$c = \sum_i \alpha_i (g_i \cdot x^* + f(g_i)) = \mathbf{0} \cdot x^* + \sum_i \alpha_i f(g_i) = \sum_i \alpha_i f(g_i).$$

$\qquad\square$

## A.2 Proofs from Section 3

**Theorem 7.** *The value of this game is $V^T = \mathbb{E}_{G \sim \mathcal{B}_T}\left[\frac{1}{2\sigma} G^2\right] = \frac{T}{2\sigma}$.*

*Proof.* Starting from Eq. (10),

$$\mathbb{E}_{G \sim \mathcal{B}_T}[G^2] = \frac{1}{2^T} \sum_{i=0}^{T} \binom{T}{i} (2i - T)^2 \qquad\qquad Eq.\ (10)$$

$$= \frac{1}{2^T}\left(4\sum_{i=0}^{T}\binom{T}{i} i^2 - 4T\sum_{i=0}^{T}\binom{T}{i} i + T^2 \sum_{i=0}^{T}\binom{T}{i}\right)$$

and since $\sum_{t=0}^{T}\binom{T}{t} = 2^T$, $\sum_{t=0}^{T}\binom{T}{t} t = T 2^{T-1}$, $\sum_{t=0}^{T}\binom{T}{t} t^2 = (T + T^2) 2^{T-2}$,

$$= \frac{1}{2^T}\left(4(T + T^2)2^{T-2} - 4T(T 2^{T-1}) + T^2 2^T\right)$$

$$= (T + T^2) - 2T^2 + T^2 = T.$$

The result then follows from linearity of expectation. $\qquad\square$

**Theorem 8.** *Consider the game where $\mathcal{G} = [-1, 1]$ with benchmark $L(G) = -\exp(G/\sqrt{T})$. Then*

$$V^T = \left(\cosh \tfrac{1}{\sqrt{T}}\right)^T \leq \sqrt{e}$$

*with the bound tight as $T \to \infty$. Let $\tau = T - t$ and $G_t = g_{1:t}$, then the conditional value of the game is $V_t(G_t) = \left(\cosh \frac{1}{\sqrt{T}}\right)^{\tau} \exp\left(\frac{G_t}{\sqrt{T}}\right)$ and the player's minimax optimal strategy is:*

$$x_{t+1} = -\exp\left(\frac{G_t}{\sqrt{T}}\right) \sinh \frac{1}{\sqrt{T}} \left(\cosh \frac{1}{\sqrt{T}}\right)^{\tau-1} \tag{12}$$

*Proof.* First, we compute the value of the game:

$$V^T = \mathop{\mathbb{E}}_{G \sim \mathcal{B}_T}\left[-L(G)\right] = 2^{-T} \sum_{i=0}^{T} \binom{T}{i} \exp\left(\frac{2i - T}{\sqrt{T}}\right)$$

$$= 2^{-T} \exp\left(-\sqrt{T}\right) \sum_{i=0}^{T} \binom{T}{i} \left(\exp\left(2/\sqrt{T}\right)\right)^i$$

$$= 2^{-T} \exp\left(-\sqrt{T}\right) \left(1 + \exp\left(2/\sqrt{T}\right)\right)^T,$$

where we have used the ordinary generating function, $\sum_{i=0}^{T} \binom{T}{i} x^i = (1 + x)^T$. Manipulating the above expression for the value of the game, we arrive at $V^T = \cosh(1/\sqrt{T})^T$. Using the series expansion for cosh leads to the upper bound $\cosh(x) \leq \exp(x^2/2)$,

from which we conclude

$$V_T = \left(\cosh\left(1/\sqrt{T}\right)\right)^T \leq \exp\left(\frac{1}{2T}\right)^T = \sqrt{e}.$$

Using similar techniques, we can derive the conditional value of the game, letting $\tau = T - t$ be the number of rounds left to be played:

$$V_t(G_t) = 2^{-\tau} \sum_{i=0}^{\tau} \binom{\tau}{i} \exp\left(\frac{G_t + 2i - \tau}{\sqrt{T}}\right) = 2^{-\tau} \exp\left(\frac{G_t - \tau}{\sqrt{T}}\right) \left(1 + \exp\left(2/\sqrt{T}\right)\right)^{\tau}.$$

Following Eq. (9) and simplifying leads to the update of Eq. (12). It remains to show $\lim_{T\to\infty} V_T = \sqrt{e}$. Using the change of variable $x = 1/\sqrt{T}$, equivalently we have $\lim_{x\to 0} \cosh(x)^{\frac{1}{x^2}}$. Examining the log of this function,

$$\lim_{x\to 0} \log\left(\cosh(x)^{\frac{1}{x^2}}\right) = \lim_{x\to 0} \frac{1}{x^2} \log \cosh(x) = \lim_{x\to 0} \frac{1}{x^2}\left(\frac{x^2}{2} - \frac{x^4}{12} + \frac{x^6}{45} - \frac{17x^8}{2520} + \dots\right) = \frac{1}{2},$$

where we have taken the Maclaurin series of $\log \cosh(x)$. Using the continuity of exp, we have against any adversary,

$$\lim_{x\to 0}\left(\cosh(x)^{\frac{1}{x^2}}\right) = \exp\left(\lim_{x\to 0} \log\left(\cosh(x)^{\frac{1}{x^2}}\right)\right) = \sqrt{e}.$$

$\square$

**Theorem 9.** *Consider a one dimensional game with $\mathcal{G} = [-1, 1]$ with benchmark function $L$ non-positive on $\mathcal{G}^T$. Then for the optimal betting strategy we have that $|x_t| \leq -\sum_{s=1}^{t} g_s x_s + V^T$, and further $V^T \geq \sum_{s=1}^{t} g_s x_s$ for any $t$ and any sequence $g_1, \dots, g_t$.*

*Proof.* We need to prove

$$\sum_{s=1}^{t} g_s x_s \leq V^T \tag{15}$$

and

$$|x_t| \leq -\sum_{s=1}^{t} g_s x_s + V^T. \tag{16}$$

The definition of the value of the game and the fact the algorithm is minimax optimal ensures

$$\sum_{t=1}^{T} g_t x_t - L(G) \leq V^T$$

or, since $-L(G) \geq 0$,

$$\sum_{t=1}^{T} g_t x_t \leq V^T. \tag{17}$$

Now, suppose on some round $t$ we have $\sum_{s=1}^{t} g_s x_s > V^T$. Then, the adversary can simply play $g_\tau = 0$ for rounds $t+1, \ldots, T$, which implies

$$\sum_{s=1}^{T} g_s x_s = \sum_{s=1}^{t} g_s x_s > V^T,$$

contradicting Eq. (17). Hence, Eq. (15) must hold. Further, if the player ever chose a bet so large it violated Eq. (16), the adversary could choose $g_t \in \{-1, 1\}$ in order to violate Eq. (17). $\qquad \square$

**Theorem 10.** *Consider the game between an adversary who chooses losses $g_t \in [-1, 1]$, and a player who chooses $x_t \in \mathbb{R}$. For a given sequence of plays, $x_1, g_1, x_2, g_2, \ldots, x_T, g_T$, the value to the adversary is $\sum_{t=1}^{T} g_t x_t - |g_{1:T}|$. Then, when $T$ is even with $T = 2M$, the minimax value of this game is given by*

$$V_T = 2^{-T} \frac{2M \, T!}{(T-M)! M!} \leq \sqrt{\frac{2T}{\pi}}.$$

*Further, as $T \to \infty$, $V_T \to \sqrt{\frac{2T}{\pi}}$. Let $B$ be a random variable drawn from $\mathcal{B}_{T-t}$. Then the minimax optimal strategy for the player given the adversary has played $G_t = g_{1:t}$ is given by*

$$x_{t+1} = \Pr(B < -G_t) - \Pr(B > -G_t) = 1 - 2\Pr(B > -G_t) \in [-1, 1]. \tag{14}$$

*Proof.* Letting $T = 2M$ and working from Eq. (10),

$$V^T = -\mathop{\mathbb{E}}_{G \sim \mathcal{B}_T}[L(G)] = \frac{2}{2^T} \sum_{i=0}^{T} \binom{T}{i} |i - M| = \frac{2M}{2^T} \binom{2M}{M} = 2^{-T} \frac{2M \, T!}{(T-M)! M!}, \tag{18}$$

where we have applied a classic formula of de Moivre [1718] for the mean absolute deviation of the binomial distribution (see also Diaconis and Zabell [1991]). Using a standard bound on the central binomial coefficient (based on Stirling's formula),

$$\binom{2M}{M} = \frac{4^M}{\sqrt{\pi M}} \left(1 - \frac{c_M}{M}\right) \tag{19}$$

where $\frac{1}{9} < c_M < \frac{1}{8}$ for all $M \geq 1$, we have

$$V^T \leq 2M \frac{1}{\sqrt{\pi M}} = \sqrt{\frac{2T}{\pi}}.$$

As implied by Eq. (19), this inequality quickly becomes tight as $T \to \infty$.

In order to compute the minimax algorithm, we would like a closed form for $V_t(G_t) = -\mathbb{E}_{G^\tau \sim \mathcal{B}_\tau}\left[L(G_t + G^\tau)\right]$, where $G_t = g_{1:t}$ is the sum of the gradients so far, $\tau = T - t$ is the number of rounds to go, and and $G^\tau = g_{t+1:T}$ is a random variable giving the sum of the remaining gradients. Unfortunately, the structure of the binomial coefficients exploited in the proof of Theorem 10 does not apply given an arbitrary offset $G_t$. Nevertheless, we will be able to derive a formula for the update that is readily computable. Letting $B$ be a random variable with distribution $\mathcal{B}_\tau$, the update of Eq. (9) becomes

$$x_{t+1} = \frac{1}{2} \sum_{b=-\tau}^{\tau} \Pr(B = b) \Big( |G_t + b - 1| - |G_t + b + 1| \Big).$$

Whenever $G_t + b \geq 1$, the difference in absolute values is $-2$, and whenever $G_t + b \leq 1$, the difference is $2$. When $G_t + b = 0$, the difference is zero. Thus,

$$x_{t+1} = \frac{1}{2} \left( \Pr(B > -G_t)(-2) + \Pr(B < -G_t)(2) \right) = \Pr(B < -G_t) - \Pr(B > -G_t).$$

$\qquad \square$

# B  A Symmetric Betting Algorithm

The one-sided algorithm of Theorem 8 has

$$\text{Loss} = V^T + L(G) \leq -\exp\left(\frac{G}{\sqrt{T}}\right) + \sqrt{e}.$$

In order to do well when $g_{1:T}$ is large and negative, we can run a copy of the algorithm on $-g_1, \ldots, -g_T$, switching the signs of each $x_t$ it suggests. The combined algorithm then satisfies

$$\text{Loss} \leq -\exp\left(\frac{G}{\sqrt{T}}\right) - \exp\left(\frac{-G}{\sqrt{T}}\right) + 2\sqrt{e}$$

$$\leq -\exp\left(\frac{|G|}{\sqrt{T}}\right) + 2\sqrt{e},$$

and so following Eq. (13) and Theorem 1 of Streeter and McMahan [2012], we obtain the desired regret bounds. The following theorem implies the symmetric algorithm is in fact minimax optimal with respect to the combined benchmark

$$L_C(G) = -\exp\left(\frac{G}{\sqrt{T}}\right) - \exp\left(\frac{-G}{\sqrt{T}}\right).$$

**Theorem 16.** *Consider two 1-D games where the adversary plays from $[-1, 1]$, defined by concave functions $L_1$ and $L_2$ respectively. Let $x_t^1$ and $x_t^2$ be minimax-optimal plays for $L_1$ and $L_2$ respectively, given that $g_1, \ldots g_{t-1}$ have been played so far in both games. Then $x_1 + x_2$ is also minimax optimal for the combined game that uses the benchmark $L_C(G) = L_1(G) + L_2(G)$.*

*Proof.* First, taking $\tau = T - t$ and using Theorem 4 three times, we have

$$V^C(g_1, \ldots, g_t) = -\mathop{\mathbb{E}}_{G^\tau \sim \mathcal{B}_\tau}\left[L_1(g_{1:t} + G^\tau) + L_2(g_{1:t} + G^\tau)\right]$$

$$= -\mathop{\mathbb{E}}_{G^\tau \sim \mathcal{B}_\tau}\left[L_1(g_{1:t} + G^\tau)\right] - \mathop{\mathbb{E}}_{G^\tau \sim \mathcal{B}_\tau}\left[L_2(g_{1:t} + G^\tau)\right]$$

$$= V^1(g_1, \ldots, g_t) + V^2(g_1, \ldots, g_t),$$

using linearity of expectation. Then, using Eq. (9) for each of the three games, we have

$$x_t^C = \arg\min_x \max_g gx + V_C(g_1, \ldots, g_{t-1}, g)$$

$$= \frac{1}{2}\left(V_C(g_1, \ldots, g_{t-1}, -1) - V_C(g_1, \ldots, g_{t-1}, +1)\right)$$

$$= \frac{1}{2}\left(V_1(g_1, \ldots, g_{t-1}, -1) + V_2(g_1, \ldots, g_{t-1}, -1) - V_1(g_1, \ldots, g_{t-1,1}) - V_2(g_1, \ldots, g_{t-1}, +1)\right)$$

$$= x_t^1 + x_t^2.$$

$\square$