[Reviews · NeurIPS 2013]

Submitted by Assigned_Reviewer_4

SUMMARY AND RELATED WORKS

This paper is about online linear optimization with an unconstrained comparator set (e.g., the whole R^n space). Instead of trying to be competitive with the best linear predictor x \in R^n, the aim is typically to be competitive with the best penalized comparison vector. In this setting, and assuming that the convex hull of the set of moves of the adversary is a full-rank polytope in R^n, the authors provide stochastic techniques to derive the value of the game, as well as the optimal strategies for the player and the adversary. These abstract results are then applied to the one-dimensional case (a straightforward extension to arbitrary dimension with coordinate-decomposable games is also provided, but no thorough application was worked out). Three 1-dimensional examples are studied in details: linear optimization with a quadratic regularizer, non-stochastic betting, and the standard online linear optimization problem on [-1,1] (where the authors recover known results from their new general technique).

This paper is a continuation of a series of papers that addressed adversarial online learning problems from a stochastic viewpoint (Abernethy et al. COLT 2008, Abernethy et al. COLT 2009, Rakhlin et al. NIPS 2012). It extends those works to the case of unconstrained comparator sets. Note however that unconstrained online convex optimization is definitely not new: papers like Duchi et al. (Composite Objective Mirror Descent, COLT 2010) already provided regret upper bounds for unconstrained comparison vectors. The novelty here is that this setting is studied from an exact minimax viewpoint.

OVERALL ASSESSMENT

My overall opinion about the paper is quite positive for the following reasons:

PROS

1) The paper is well written and pleasant to read (all necessary intuitions are provided when needed). Mathematics seem correct.
2) Though not ground-breaking, this paper makes a further step towards the understanding of unconstrained linear optimization.
3) The application to non-stochastic betting is really nice, with a tight analysis and a simple algorithm.

CONS

4) There are not sufficiently many applications of the abstract results: Theorems 7 and 9 are nice for a warm-up, Theorem 8 is interesting, but I would have also expected a study of other forms of penalized regret like the ones already encountered with online mirror descent. In particular, can the techniques of the present paper be adapted to multi-dimensional games that are not coordinate-decomposable? Most Bregman-divergence-based regularizations are indeed of this form.
5) Contrary to what is announced on line 326, it seems that the constraint that the player cannot bet more than what he owns is not imposed to the game. I understand that it is possible to make sure that he loses no more than 1$ at the end of the game, but what can be said on the whole path from t=1 to t=T?

OTHER COMMENTS
6) Related Work (Page 2): references about unconstrained regret upper bounds with Online Mirror Descent are missing.
7) Lines 105-107: the proposed 'alternative' is not new: actually, the comparison of the loss to the best penalized loss is known in statistics as an 'oracle inequality' from quite a long time (see, e.g., the references in "Concentration Inequalities and Model Selection", Massart, 2007). Parallely, in the machine learning community, some works like the one of Vovk ("Competitive on-line statistics", 2001) already began to formalize such ideas in the previous decade.
8) Line 141: hull(G)=hull(G')
9) Line 206: is the Lipschitz assumption really needed? Is boundedness sufficient?
10) Statement of Theorem 3: I suggest to move the fact that [we can find a full-rank set of corners {g_1,...,g_{n+1}} of the polytope hull(G)] before the statement of Theorem 3. This is indeed straightforward in all subsequent applications. (Furthermore, the term 'distinct' points is too weak, full-rank would be more precise).
11) Line 262: it is implicitely assumed that V_{t+1} is Lipschitz (or bounded? -> see above).
12) Line 288: the random variable B was not defined earlier.
13) Theorem 7: recall that g_t \in [-1,1].
14) Title of Section 3.2: what does "all feasible sets" stand for?
15) Line 341: please explain "bet everything every round ... 2^T"
16) Theorem 7 and the equation on line 683 are actually straightforward by writing that B is the sum of independent Rademacher variables.
17) Theorem 8: I think that the first exponential contains (G_t-\tau)/sqrt{T} (no "-1" term), and that the exponent of the last term is \tau-1 instead of \tau (the reason is that the authors used V_t instead of V_{t+1}).
18) Several typos : argmin -> min (line 117), that that -> that (line 121), Theorems 1 and 2 -> Corollary 2 (line 244), V_t -> V_{t+1} (line 286), g_{1:T} -> g_{1:t} (lines 288 and 289), -\partial L -> \partial L (line 292), *2* B sqrt{T} (line 338 + cf consequences in the following lines), |G| \leq sqrt{T} (line 355), loss functions -> losses (line 411)


ADDITIONAL COMMENT AFTER THE AUTHORS' REBUTTAL

I thank the authors for their detailed answer. It seems that point 5) can be taken care of. However, I regret that the pratical implications of the abstract results are too narrow in the submitted version, and that the amount of work needed to address the (interesting) applications mentioned in the rebuttal is too large. The proposed modified paper should have clear chances of acceptation in a later conference.
Summary: This paper makes an original further step towards the understanding of unconstrained linear optimization; intuitions are well-explained and make the paper pleasant to read. However, the practical implications of the abstract results are a little too narrow at the moment, and I fear that the revision outlined in the rebuttal will require a too large amount of work.

Submitted by Assigned_Reviewer_5

The paper theoretically examines online linear optimization games where the optimizing player is minimizing regret with respect to some benchmark loss. They give a recursive characterization of the value of the game and use this characterization to derive the minimax optimal strategies.

This paper is interesting and dense. Very dense.

I must admit that I'm not an active researcher in this area, but I am an active follower and "consumer" of such results. To the paper's credit I was able to follow (although not entirely verify) much of the paper's derivation, and found much of the analysis novel (to me) and very clever. The result for the betting game was particularly interesting.

However, I was frequently lost on the main thrust of paper's contributions. By that I don't mean the theoeretical contributions in terms of theorems proven, but rather the meaning or implications of the theoretical contributions. As a community, I think it's important for NIPS not to fragment; and to avoid this we need to be careful that our papers aren't only understood by the handful of people working in an area. This means providing more context to the theoretical results and why they are important and their implications for the broader machine learning community. The place for such context in this paper is Section 3, where the implications are drawn for three different settings, but these settings are simply too abstract, and now the question is what are the implications of these results. Maybe this is where a discussion section could greatly improve the paper. However, the paper abrubtly ends in Section 3, almost as if there are pages missing (maybe there are).

I think this paper could be a very strong paper (top 5% of accepted papers strong), but it unfortunately misses that opportunity. It might miss it so far as to not warrant publication.

Minor Comments:

066: "We this in mind" => "With this in mind"
076: "via a penalty functions" => "via a penalty function"
305: "cannot have no a regret" => "cannot have a no regret"
Summary: The paper makes some very interesting theoretical contributions for online linear optimization games. The authors, though, fail to give context or implication for these results.

Submitted by Assigned_Reviewer_7

The paper presents an analysis of the minimax formulation of a repeated decision-making task as a sequential zero-sum game. Unlike the usual setup where the players are restricted to choosing a strategy in the simplex over a predetermined set of choices, the players in this paper are generally unconstrained. The authors characterize the game using martingale sequences and show that minimax-optimal values can be expressed using discrete gradients over the values of the subgames. The authors then propose minimax-optimal algorithms under alternative notions of benchmark loss for a subset of games of this form call "coordinate-decomposable" games.

The paper presents a novel characterization of repeated unconstrained optimization problems as well as several new theoretical results for these problems. The first is formulation of the game value as the supremum of the expected value as a martingale sequence. The second is that optimal strategies can be expressed using the discrete gradient of the linear interpolation of pairs of opponent strategies and values of each subgame at time step t+1. Some mild assumptions are required, but both of these results are significant contributions and appear to be novel. Furthermore, the authors define new regret notions for an unconstrained betting game and derive minimax-optimal strategies that have a non-conventional form.

The paper is dense in some parts but overall the presentation is clear given the content with some minor exceptions. There is no conclusion; the four bullet points in the introduction should be summarized by a single sentence each and then the elaborate explanations of each moved to a conclusion that summarizes the contributions of the paper. The explanation of the betting game itself and the reasons that the previous benchmarks are unsatisfactory should be made more clear.

As this is not my area of expertise, some of the proof steps were difficult to get through. The paper can be made generally more accessible by more pointers to existing literature.

Some of the results resemble those from two-person zero-sum extensive game theory and the connections may be worth mentioning. For example, some steps of the proof of Theorem 1 seem similar to those of the simplex algorithm used in solving LP formulations of zero-sum games. The construction of Theorem 3 and setting f(g_t) = V_t(...) in equation 8 looks similar to backward induction. Are the concepts in this paper generalizations of these concepts in the same way that Sion's Minimax Theorem generalizes von Neumann's minimax theorem?

Some specific comments:

- Eq 3: should argmin be min? And why the change from inf? Similarly at the top of page 2: L(G) is a value, so should it be min vs. argmin?

- In Figure 1, why use a circle over x? This notation is inconsistent with the rest of the text in the main paper.

- Page 4, provide a citation for Sion's Minimax theorem.

- Bottom of Page 4 and again several times in the appendix: the set of pairs {(g^1, -f(g^1), ... } is missing a closing parenthesis.


Summary: The paper presents several new theoretical results for minimax formulations of repeated unconstrained optimization problems. The contributions are significant and the presentation is clear.

Submitted by Assigned_Reviewer_8

This paper studies minimax algorithms for online linear optimization settings (in the d-dimensional Euclidean space) where the linear loss of the comparator x is increased by a penalization term Psi(x). This allows the regret to be computed against any comparator x, rather than only comparators belonging to some constrained region. A (non-constructive) characterization of the minimax value is derived in terms of a martingale process under the assumption that the loss vectors belong to a set whose convex hull is a full-rank polytope. Explicit minimax algorithms are constructed for special setting (coordinate-decomposable games, 1-dimensional games with quadratic penalties, betting games).

The paper is well written and technically solid. However, the significance of the problem is unclear. The settings considered in the paper are quite specific, and the impact of results and proof techniques on more general settings remains also unclear. The scope appears too limited to provide useful insights. I think the work is potentially promising but too preliminary - I would like to see applications to more practical settings.

POST-REBUTTAL ADDENDUM =============================

I thank the authors for explaining the range of applications of their results. Still, it looks like the paper needs a substantial amount of revision.
Summary: A potentially interesting paper but limited in scope and impact.
Author Feedback

Author rebuttal: We would like to sincerely thank the reviewers for their careful reading of the paper, and their useful feedback. We were pleased at the positive feedback provided.

Several reviewers suggested that our results might be too narrow to be of broad interest, and would have little useful application. Looking back at the submitted paper, it’s obvious we should have been more clear in highlighting the applicability and scope of our contributions. While we leave empirical evaluation to future work, we believe there is a clear path forward here that will be very relevant to the NIPS community. Despite appearances, the setting and results we present are anything but niche: the problem framework of “unconstrained online optimization” is a fundamental template for (and strongly motivated by) several online learning settings, and the results we develop are very applicable to a wide range of commonly studied algorithmic problems. It is a pity we did not make this significantly clearer, and it is an error we intend to correct.

Let us give a sketch of what we shall add to the paper with some motivating examples. First, the classic algorithm for linear pattern recognition, the Perceptron, can be seen as an algorithm for unconstrained linear optimization. Going further, methods for training a linear SVM or a logistic regression model, such as stochastic gradient descent or the Pegasos algorithm, are unconstrained optimization algorithms. Finally, there has been recent work in the pricing of options and other financial derivatives (see Abernethy et al, and some earlier work by DeMarzo, Kremer, and Mansour) that can be described exactly in terms of a repeated game which fits nicely into our framework.

We also wish to emphasize that the algorithm of Sec 3.2 is both practical and easily implementable: for a multi-dimensional problem one needs to only track the sum of gradients for each coordinate (similar to Regularized Dual Averaging), and compute Eq. (14) for each coordinate to derive the appropriate strategy (for numerical reasons, computing (14) in log space and then exponentiating will be necessary). We emphasized this algorithm as a tool for making a-priori unconstrained bets/investments. However, as [1] shows, the bounds we give for this minimax-optimal algorithm yield bounds on standard regret that are superior to those of classic algorithms (exponentiated gradient descent, online gradient descent and mirror descent) when the norm of the optimal point is not known in advance. Thus, we have offered a quite novel algorithm for a fundamental problem that is demonstrably quite different than standard approaches, and have proved it can offer much better bounds. It is clear that we need to make this more visible in the final version.

Reviewer 4: To point 4), the application to non-decomposable games in higher dimensions is a natural next step, and one we hope to address in future work. Point 5) is a good one, we will provide a more clear argument in the revised version. To sketch the argument, suppose on some round the player bets more than his current balance. Then, the adversary can play +1 or -1 as required to ensure the player loses this bet (dropping the player below the allowed floor on loss). Then, the adversary can play 0 gradients for the remainder of the game to lock in this loss. Thus, the claims of line 326 must hold if the algorithm of Thm 8 is followed (with an appropriate scale factor). We appreciate the reviewer’s careful reading of the paper, and will of course address the points made under OTHER COMMENTS (including the additional citations).

Reviewer 5: There is indeed a high-level connection to extensive-form games. We will consider whether this can be made more precise. Eq. (3) should indeed be a min; the change from inf to min is simply to clarify the argmin must exist. We appreciate the other specific comments, we'll fix these in the final version.

[1] “No-regret algorithms for unconstrained online convex optimization.” In NIPS, 2012. Streeter and McMahan.